# The Free Energy Principle: Good Science and Questionable Philosophy in a Grand Unifying Theory

**DOI:** 10.3390/e23020238

**Published:** 2021-02-19

**Authors:** Javier Sánchez-Cañizares

**Affiliations:** Mind-Brain Group, Institute for Culture and Society, University of Navarra, 31009 Pamplona, Spain; js.canizares@unav.es

**Keywords:** Free Energy Principle, Markovian Monism, grand unification theory, complex adaptive systems, Bayesian brain, principle of individuation

## Abstract

The Free Energy Principle (FEP) is currently one of the most promising frameworks with which to address a unified explanation of life-related phenomena. With powerful formalism that embeds a small set of assumptions, it purports to deal with complex adaptive dynamics ranging from barely unicellular organisms to complex cultural manifestations. The FEP has received increased attention in disciplines that study life, including some critique regarding its overall explanatory power and its true potential as a grand unifying theory (GUT). Recently, FEP theorists presented a contribution with the main tenets of their framework, together with possible philosophical interpretations, which lean towards so-called Markovian Monism (MM). The present paper assumes some of the abovementioned critiques, rejects the arguments advanced to invalidate the FEP’s potential to be a GUT, and overcomes criticism thereof by reviewing FEP theorists’ newly minted metaphysical commitment, namely MM. Specifically, it shows that this philosophical interpretation of the FEP argues circularly and only delivers what it initially assumes, i.e., a dual information geometry that allegedly explains epistemic access to the world based on prior dual assumptions. The origin of this circularity can be traced back to a physical description contingent on relative system-environment separation. However, the FEP itself is not committed to MM, and as a scientific theory it delivers more than what it assumes, serving as a heuristic unification principle that provides epistemic advancement for the life sciences.

## 1. Introduction

The Free Energy Principle (FEP) inspires one of the most comprehensive frameworks for the study of complex adaptive systems. Originated by Karl Friston and colleagues, it spans active research in several life science fields under the overarching principle of internal free energy minimization. For this reason, the FEP can be counted among contemporary attempts to understand complex systems through variational principles—see e.g., [1,2,3,4] for a short review. However, the FEP seems to encompass a broader picture because of its alleged ability to include more restricted approaches in neurosciences, including the mind-brain problem ([5], p. 136). What is more, some scientific commentators claim it might hold the key to artificial intelligence [6].

Throughout the last two decades, FEP formalism has been presented with technical aspects that may prove threatening for the uninitiated. Admittedly, changes to notation and symbols can misguide readers, hence FEP papers typically include tables with mathematical definitions of key concepts and figures that help visualize the underlying schema. Nevertheless, the theory increasingly appears as a promising and encompassing framework to study living systems, both dia- and syn-chronically, and has already been considered in the literature as a grand unifying theory (GUT) ([7], pp. 5–7) for the understanding of evolution, sentience and consciousness, as well as of human cognition and the acquisition of culture [8]. Some have even argued that the FEP introduces an unresolved tension between the secluded organism and its necessary openness and interaction with the environment, which finds its philosophical home in Hegel’s dialectics [9].

During the last decade, criticism of the FEP has emerged, see e.g., [10]. They cannot but be welcomed inasmuch as the FEP’s initial assumptions and philosophical consequences require clarifications and discussion of their drawbacks. Recently, however, the FEP has been attacked on more general grounds—targeted for its distinction as a possible GUT. Whereas such criticism might be partially justified in terms of the FEP’s failure to achieve its goal as it stands, it might also go too far and ruin the main thrust of scientific reduction. Hence the existence of apparently irreducible models to explain, for instance, the activity of mesocorticolimbic dopaminergic systems [7] need not necessarily count as evidence against the FEP’s aspirations; on the contrary, it should stir up better explanations through proper modeling within the FEP framework.

This paper assumes the FEP’s implicit claim of epistemic reduction in its understanding of complex adaptive systems and concentrates on reviewing its explanatory power. Beyond technical criticisms regarding the emergence of Bayesian inference via Markov blankets and the FEP’s universality [11], Colombo and Wright [12] tried to clarify how the FEP relates to the two most prominent theoretical approaches to life science phenomena, namely organicism and mechanism. Notwithstanding, Friston and colleagues have recently published a discussion of the FEP’s compatibility with different philosophical perspectives, ranging from monism to dualism [13], and explicitly adopting what they dub Markovian Monism (MM). This move naturally demands careful review of the philosophical assumptions behind the FEP’s main concepts and tenets, as well as its links (or lack thereof) to key philosophical concepts like representation, supervenience and individuation.

Should the FEP’s newly minted ontology be understood as a response to general criticism of the FEP as a GUT? The present work maintains both that foregoing critiques—especially those by Colombo and Wright [12]—can be rejected on grounds other than MM and that the latter is a misleading basis for the FEP framework due to its circular reasoning. After introducing the FEP’s main concepts and assumptions in non-technical jargon in Section 2, this paper endeavors to (1) distinguish between misplaced and valuable criticisms in Colombo and Wright’s review of the FEP by defending its role as a scientific theory (Section 3 and Section 4) and (2) assess the value of MM as an interpretation of the FEP. Section 5 distils the most relevant ideas in MM, Section 6 makes explicit some of MM’s assumptions regarding key philosophical concepts, and Section 7 begs the fundamental question of extant individual systems in nature. Ultimately, the FEP may withstand criticism inasmuch as it is not committed to MM for two main reasons: First, as the FEP’s philosophical interpretation, MM obtains what it assumes, i.e., a dual information geometry that allegedly explains epistemic duality because of its prior, initial dual assumptions. Second, such circular reasoning stems from the FEP’s implicit reliance on a non-fundamental, relative system-environment separation. The conclusions herein explain why this double circularity proves harmless for the FEP inasmuch as it remains a principle limited to science, providing epistemic advancement to life sciences. Whereas the FEP as a scientific theory can be based on such circularities, the FEP’s philosophical interpretations, like MM, cannot. Science needs philosophical assumptions through which its theories can be linked with observations but does not need to justify them if they remain open to philosophical criticism. However, philosophical interpretations of scientific theories that aim at ultimate foundations may not contain unresolved circular reasoning. That is why MM is a questionable philosophical backdrop for the FEP.

## 2. The Main Assumptions of FEP Formalism

FEP formalism has been developed in several contributions in keeping with different emphases [5,8,13,14,15,16,17,18,19,20]. By far, [21] provide the clearest introduction to this formalism and its mathematical assumptions, with the meaning of symbols and equations carefully spelled out.

### 2.1. The Markov Blanket at the Core of the System/Environment Distinction

FEP defenders begin their description of systems of interest with the introduction of Markov blankets as key to enabling a workable distinction between the system and the environment. Even if not controversial in usual scientific activity, distinguishing between system and environment is non-trivial in, e.g., quantum mechanics and entails some relativity due to the observer’s involvement [22,23]. This methodological separation of domains obeys practical criteria—including ranges of energy [24]—that are founded on weak grounds.

The conceptual introduction of a Markov blanket not only allows for a well-defined border between the system and the environment, but also permits the description of a network of conditional interdependencies between the system, its Markov blanket, and the environment. This allows for a useful mathematical description of the system and its overall dynamics, both internal and as response to the environment. Within this basic framework, it becomes possible to identify life-related phenomena as learning or cycles of perception and action—see e.g., ([5], p. 128). For the sake of specificity and following the partition of [13], one may consider the following relevant sets of states or degrees of freedom (I consider both concepts equivalent throughout the paper) in the FEP framework: “external degrees of freedom” $=\left\{\eta_{i}\right\}$, with i=1, …, I; “internal degrees of freedom” = $\left\{\mu_{j}\right\}$, with j=1, …, J; “blanket degrees of freedom” $=\left\{s_{k}\right\}\;U\;\left\{a_{l}\right\}$, with k=1, …, K; l=1, …, L; where $\left\{s_{k}\right\}$ are “sensory degrees of freedom” and $\left\{a_{l}\right\}$ are “active degrees of freedom”. Typically, I≫J>K,L, which simply assumes that the environment has many more possible degrees of freedom than the system and its internal degrees of freedom are usually more than its sensory and active states.

The advantage of such factorization is that, in terms of probability distributions, a Markov blanket renders internal states conditionally independent of external states. Additionally, sensory states are not influenced by internal states and active states are not influenced by external states [13]. In other words, system states’ initially obvious dependence on environmental states is parametrized in such a way that it becomes mediated by blanket degrees of freedom alone. The system always senses and acts on the world through mediation of its own Markov blanket; its sensory states directly (By “directly”, I mean here dependence at the same time. Obviously, across the system’s history, there are many deferred dependencies (occurring at later times). Yet, formalism allows us to concentrate on synchronic dependence.) depend on environmental states and its active states depend on the system’s internal states, but not on environmental states. The former describes how the environment impinges on the system and the latter how the system deals with the environment. Factorization of dependencies between degrees of freedom defines the system within the FEP framework.

### 2.2. Featuring System Dynamics: A Non-Equilibrium Steady State

Since blanket states also belong in system characterization and partake of system dynamics, one may still consider the whole set of system or *particular* states as $x=\left\{\mu_{j}\right\}\;U\;\left\{s_{k}\right\}\;U\;\left\{a_{l}\right\}$—see Figure 1 in [13]. Moreover, as long as a system does exist with its own featuring states, one may safely assume Langevin dynamics to describe the change in the system’s degrees of freedom over time: dx/dt=f(x,t)+ω, where x represents the whole set of the system’s degrees of freedom, f(x,t) is a (hopefully) smooth function of time and the degrees of freedom only and ω stands for random fluctuations. A further assumption of the FEP framework is that the system always finds itself in a steady state of non-equilibrium. This feature is a benchmark of complex adaptive systems but remains controversial when examining if it suffices to encompass the whole system’s timespan. I will come back to this issue in Section 7. For now, the assumption is that, given such factorization (external, internal, sensory, and active states of a biological system) and their relationships of conditional dependence or independence, one may just concentrate on the system dynamics that make physical states be “confined to a bounded subset of states [an attractor] and remain there indefinitely” ([12,16], p. 2106).

The FEP framework thus assigns a probability distribution for the particular states, p(x), and assumes the existence of an attractor in some region of the system’s phase space; this attractor defines the system’s most probable states. In the steady state regime, the probability of states within the attractor does not change over time even though the system is far from thermodynamic equilibrium—resisting increasing entropy in the environment whilst keeping its internal entropy at bay. p(x) must then satisfy the conditions of a steady state solution for the Fokker Planck equation—see Figure 2 in [13] for details. If one demands dp/dt=0, the solutions point to, on average, states of any system with an attracting set conforming to a gradient flow on surprisal, establishing a “lawful relationship between the flow of states at any point in state space and the probability density”.

When dealing with the different terms of the steady state solution, there is some freedom in how to identify them. For instance, in keeping with the factorization of particular states as internal and blanket degrees of freedom, “the mechanics of internal and active states can be regarded as perception and action, where both are in the service of minimizing a particular surprisal” ([13], n. 7). Again, beyond technicalities, it is important to note that (1) identifications crucially depend on the Markov blanket partition and the division between system and environment, (2) since sensory states do not directly depend on internal states, they are not degrees of freedom influencing the gradient flow setting up the steady state. Such influence is the prerogative of internal $\left\{\mu_{j}\right\}$ and active states $\left\{a_{l}\right\}$. The former can arguably be equated to “perception” and the latter to “action”. Hence, the formalism built assuming a Markov blanket, Langevin dynamics, and a steady state solution for the probability density of particular states could serve to describe, explain and understand the whole range of phenomena related to complex adaptive systems.

### 2.3. Probability Distribution “of” and “about” Things

One key tenet of the FEP framework outlined here refers to the distinction made between the probability distribution of things and the probability distribution about things. This is because the system must hold “beliefs about external states that are parametrized, represented, encoded or coherent with internal states”. What facts support such a bold claim? Basically the fact that, thanks to factorization via the Markov blanket—which makes external and internal states directly independent—there must be a 1:1 relationship between a system’s average internal state and a probability density over its external states: “The mapping between the expected (i.e., average) internal state (for any given blanket state) and a conditional density over external states (i.e., a Bayesian belief about external states) inherits from the conditional independencies that define a Markov blanket” [13]. In its full splendor, this is the benefit of interacting with and representing the external world via Markov blankets.

Turning back to probability distribution p(x), one is thus entitled to consider the system’s internal states conditioned by its Markov blanket, i.e., $p\left(x\right)=p\left(\left\{\mu_{j}\right\}|\left\{s_{k}\right\},\left\{a_{l}\right\}\right)$ in such a way that, by design, such conditioning determines how the system experiences the outer world. External states of affairs can only be represented probabilistically in a way that depends upon the Markov blanket’s dimensionality, namely K+L. The FEP framework thus associates beliefs with the probability density that is parametrized by (expected) internal states. The probability density over external states $q\left(\left\{\eta_{i}\right\}\right)$ updated by the system with internal states $\left\{\mu_{j}\right\}$ is now featured by a probabilistic belief that is parametrized by said internal states: $p\left(\left\{\mu_{j}\right\}|\left\{s_{k}\right\},\left\{a_{l}\right\}\right)$ is the system’s best guess about $q\left(\left\{\eta_{i}\right\}\right)$, and is obviously conditioned by the system’s internal states fulfilling the steady state non-equilibrium condition and its blanket states.

We see now that, in keeping with formalism and its interpretive possibilities, internal states stand in not just for themselves, but also represent what the system believes about the outer world. In other words, the probability distribution p(x) is a probability distribution of internal states and of beliefs about external states, see Equation (4) in [13]. Consequently, were someone to define a measure of distance between different probabilities in probability space and obtain a probability geometry, there is a unique geometry in some belief space that can be associated with the system’s internal (physical) state. Within the same formalism and for probability distribution, one may consider an intrinsic or extrinsic geometry that is contingent upon measuring distances between probabilities of internal (physical) states or measuring distances between probabilities of beliefs (about external states), respectively. (Whereas description of the system in terms of trajectories in phase space over time defines its “intrinsic geometry”, “extrinsic geometry” is not in a phase space of states, but rather related to a statistical manifold. A statistical manifold is a space with coordinates that are the sufficient statistics of families of probability densities or distributions; sufficient statistics are the numbers that define a given probability distribution. This dual or conjugate aspect of information geometry is more specifically tackled in Section 5.1) The powerful idea behind this is that by assuming that a system does exist, with the help of the Markov blanket factorization, one naturally obtains “beliefs” as internal representations of the external world.

### 2.4. Free-Energy Minimization as Bayesian Dynamics: Two Sides of the Same Coin

If we turn back to system dynamics, as described in Section 2.2 the gradient flows that outline the dynamics of particular states can be interpreted as a gradient flow on a variational free energy functional, i.e., a function of how beliefs are encoded in internal states and of active and sensory states as well. The FEP affirms that, so long as a Markov blanket is in play, gradient flow on variational free energy in phase space over time is equivalent to gradient flow on variational free energy in a statistical manifold, where the sufficient statistics relate to beliefs or probability distributions about external states. Actually, things become a bit more complex because free energy is shown to be an upper bound to surprisal—see, e.g., ([25], pp. 425–426) or Equation (5) in [13]. This is because free energy can be decomposed as the sum of surprisal and the Kullback-Leibler divergence between two density distributions: one of them regarding the probability of external states conditioned by blanket states; the other one encoding beliefs about the probability of external states in the system’s internal states. The Kullback-Leibler divergence is non-negative and quantifies how similar the probability density encoding beliefs is to the probability density of external states conditioned by blanket states. Hence, minimizing free energy is operationally equivalent to minimizing surprisal. (The general validity of this procedure has recently been called into question [11]. More specifically, it remains controversial that encoding external states via internal states is the best way to keep steady state dynamics and Kullback-Leibler divergence between the two probability densities low.)

Since sensory states do not directly depend on the system’s internal states, it will seem like the latter are trying to minimize exactly the same quantity, namely the surprisal of states that constitute the system. Such autonomous states can be equated with action and perception seeking to minimize surprisal or its upper bound, free energy. One might also deem minimization of particular surprisal as maximization of the value of the system’s state or a sort of “generalized homeostasis”. (See [26] for a more rigorous treatment in the context of analytical mechanics. However, that the concept of value can be equated with homeostasis in living systems, especially in the context of process philosophy, remains controversial [27].) The crucial point is that, in adapting its dynamics to minimize free energy, the system can also be understood as correspondingly changing its beliefs in keeping with Bayesian rules for updating probability densities about external states whenever new evidence is available. Because active states depend upon internal states and the beliefs that they parametrize, but not upon external states, it will look as if the system is acting on the basis of its beliefs about external states [13].

All these formulations are internally consistent with frameworks like the Bayesian brain hypothesis [28]. A system’s beliefs need not be propositional, but simply should be in line with Bayesian beliefs, i.e., conditional probability distributions that are manifest in the sense of being encoded by a physical system’s internal states. The link between inference processes and minimization of cost functions, which runs through a multitude of problems in different disciplines, is made explicit in the FEP framework. This is a small wonder inasmuch as “variational free energy is always implicitly or explicitly under the hood of any inference process, ranging from simple analyses of variance through to the Bayesian brain”. Notably, “the very existence of something leads in the natural way to a whole series of optimization frameworks in the physical and life sciences that lends each a construct validity in relation to the others” [13]. A generalized version of the FEP framework, involving policies that encompass priors over outcomes and different time scales over which to minimize free energy according to the system’s activity, has also been successfully introduced [29]. Briefly put, active inference is the counterpart of FEP formalization [30,31] and might lead to allostasis as a link from generalized homeostasis to the critical self-organization of life, as well as open up new paths for modeling the emergence of consciousness [32,33,34].

## 3. The (Misplaced) Criticism of the FEP as a GUT

Before we turn to the philosophical questions raised by the FEP framework and its characterization of complex adaptive systems, it is worth devoting some space to Colombo and Wright’s critique of the FEP’s attempt to reach GUT status [7]. These authors draw on a purportedly central case, the activity of mesocorticolimbic dopaminergic (DA) systems, in which the different explanatory models for DA systems, namely anhedonia, incentive salience, and reward prediction error hypotheses, seem to vindicate explanatory pluralism and demonstrate that scientific progress in the cognitive sciences is unlikely to be associated with a single overarching GUT. Is such an inference justified?

In cognitive science, the FEP framework can be interpreted as a form of error correction via Bayesian dynamics of previous beliefs, instantiated by a bidirectional cascade of cortical processing, see e.g., [28,35,36,37]. In fact, human observers behave as optimal Bayesian observers in many ways ([38], p. 712). Each level’s basic schema sets out that, “feed-forward connections convey information about the difference between what was expected and what actually obtained—i.e., prediction error—while feedback connections convey predictions from higher processing stages to suppress prediction errors at lower levels” ([7], p. 5). With its conceptual simplicity and powerful formalism, the free energy theory lends the ultimate rationale for Bayesian cycles of perception and action, attaining the goal of unification and scientific reduction of other higher-level explanatory principles (e.g., those utilized in psychology).

Nevertheless, Colombo and Wright [7] complain about what, for the time being, is only a would-be GUT. Currently, the FEP suggests the possibility of genuine future inter-theoretic reductions of higher-level theories to the FEP framework. However, the three abovementioned models regarding the activity of mesocorticolimbic DA systems remain irreducible and, contrary to GUT expectations, show how actual scientific practice vindicates explanatory pluralism. For instance, terms like reward and value are deemed irreducible via mathematical ‘absorption’ in favor of prior beliefs. Explanatory pluralism rejects the suggestions that, for any phenomenon, there will always be exactly one single, complete, comprehensive explanation based on a single set of fundamental principles. It assumes that scientific theories co-evolve and mutually influence one another without lower-level theories supplanting higher-level theories and hypotheses. It is precisely this inter-theory competition and selection pressure that accelerates scientific progress. Hence, “[p]rogress in neuroscience is ill-served by fervently advancing a single GUT of mind/brain that attempts to solve all problems. Rather, it is more productive to focus experimental and theoretical research on some problems, and to generate a plurality of solutions that compete as local explanations and narrowly-conceived hypotheses” ([7], pp. 6–11).

My argument is that, within this scientific context, the foregoing remarks are misplaced and unfair to the FEP framework. Not only does some of this critique turn out to be controversial, like the alleged irreducibility of reward and value to the role of priors, but, more importantly, denying the legitimacy of attempts at GUTs—even if only within neurosciences—entails a narrowly-conceived view of science. There might well be simplified, higher-level models to tackle specific practical problems, but scientific activity cannot dispense with its search for greater unification without putting meaning at risk or, worse, becoming sheer empiricism. True, Colombo and Wright [7] point out relevant, specific problems that the FEP framework needs to tackle in order to keep up with its aspirations to become a GUT. Yet, FEP does not claim at this stage to be a GUT; to date, it provides theoretical neuroscience and life disciplines with a new paradigm that can, in principle, encompass otherwise partial frameworks and shallower explanatory logics, much as evolution became the universally shared framework for biology. And, of course, the FEP may become ultimately unsuccessful.

The existence of hitherto irreducible models is not a drawback, but rather stimulates the search for more complete scientific explanations, which is a deeper stimulus than underscoring the mutual benefit of theories’ pluralistic co-evolution ([7], pp. 10–11). Colombo and Wright do not justify their pluralistic approach with such difficulties more than difficulties in explaining specific transitional states could justify finishing off the evolutionary framework. Moreover, explanatory pluralism as a last-ditch epistemic strategy smacks of an attempt at unassailability and obtains little by dismissing the epistemic core of scientific explanation. Does this mean that a scientific GUT, like the FEP aims to be, is bound to be the ultimate explanation? Not necessarily since GUTs may urgently require philosophical clarifications and/or interpretations. I will address this issue in Section 5, Section 6 and Section 7 when discussing MM, but let us first identify more profound critiques of the FEP in the foregoing authors.

## 4. Valuable Critiques of the FEP

Fortunately, Colombo and Wright have other relevant and timely critiques of the FEP that deserve closer inspection [12]. They ask for the detailed philosophical review in Section 5, Section 6 and Section 7 and require clarification of the FEP’s epistemic status: “FEP’s epistemic status remains opaque, along with its exact role in biological and neuroscientific theorizing. Conspiring against its accessibility are the varying formalisms and formulations of FEP, the changing scope of application, reliance on undefined terms and stipulative definitions, and the lack of clarity in the logical structure of the reasoning leading to FEP”.

Colombo and Wright initially praise the FEP framework as a powerful attempt at blending biology and information that might also illuminate the continuity between life and mind, as the FEP applies to any biological system. Nevertheless, concerns arise due to the FEP’s apparent silence on the nervous system’s biophysical reality and its implicit commitment to some form of cognitivism, where cognition is taken for granted within a functional scheme. However, functional analyses lack explanatory power as they are “sketches of mechanisms, in which some structural aspects of a mechanistic explanation are omitted. Once the missing aspects are filled in, a functional analysis turns into a full-blown mechanistic explanation” ([39], p. 283). Accordingly, mechanists play down the FEP’s explanatory role because of its lack of specific analysis, including the biophysical details to localize each operation with its respective component part: “Phenomena studied in the life sciences should be explained by appealing to the component parts and operations of mechanisms, where a mechanism is a spatiotemporally-organized composite system producing a phenomenon” [12]; see e.g., [40,41,42] for an improved mechanistic explanation of complex adaptive systems. Briefly stated, for mechanists, efficiency and structure determine form and function. However, obviously this perspective is itself highly controversial. (Functional explanations allow emergent functions to be realizable in multiple ways in complex adaptive systems, obviously supported by microphysical states, but not necessarily reducible to the latter’s dynamics. In this sense, functional explanations may provide functions with an ontological status beyond epiphenomenalism, which reductive mechanists may see as a drawback.)

However, to avoid getting ahead of myself, here I will focus on criticism from the organicist perspective, which raises serious doubts about the ability of physics to adequately represent organisms and their behaviors. On the one hand, organicists accuse FEP theorists of too quickly ascribing FEP framework properties and tools to organisms. Do free energy, surprise, and optimization exist as biological properties in complex adaptive systems? One could bring such concerns to the general problem of representation in scientific theories, which affects topics beyond the FEP framework, although obviously exceeds the scope of this paper.

On the other hand, organicists heavily criticize the FEP’s crucial assumption about ergodicity over the system’s span of existence. This assumption might lead towards insufficient characterizations of the organism’s phenotype and defining properties. Admittedly, FEP theorists always introduce the caveat of applying formalism to well-defined, steady phases of the system’s life because, by definition, the FEP framework cannot deal with processes like death. Ergodicity “only holds over certain temporal scales for real organisms that are on a trajectory from birth to death” ([5], sec. S1). Yet, such statements seem to fall into a sort of tautology, as the FEP framework works for ergodic phases and is unable to predict more complicated life phenomena related to phase transitions. In other words, overall, life behaves non-ergodically for organicists. Organisms “live” in extended critical phase transitions [43] or on the edge of order and chaos in the region of criticality [44,45], which disavows homeostatic stability as the core feature of living systems. (One anonymous reviewer pointed out that “ergodicity is not a problem for the FEP framework because the FEP framework only assumes local ergodicity: relative to the parts that make it up, each whole looks stationary”. Obviously, such a view is highly problematic when degrees of freedom are redefined and parts of the alleged whole are redefined—what is related to the individuation problem, more on this in Section 7. Certainly, even if life might be globally non-ergodic, it is locally ergodic enough to, e.g., be considered as having locally stable properties. Nevertheless, the problem arises at critical transition points. To follow up with an analogy suggested by the reviewer, Earth might be flat enough to build skyscrapers, but not to assume that interoceanic flights follow a straight line. To put it plainly, one must be aware of accumulated error in dealing with living systems as a temporal line of stable cycles. Such error may dramatically increase when it comes to a critical point.)

If that is the case, proper description of life might be not only non-computable in a finite number of steps, but also non-algorithmic. Because of their intrinsic historicity, organisms need not possess general characteristics that allow for complete mathematically invariant representations. Up to what point can one speak of biological symmetries that have to be preserved? Certainly, one is allowed to tackle such symmetries as a good approximation to life characteristics—see e.g., [46]—but breaks in symmetry that redefine the relationships between system and environment continually occur and some contingency beyond general principles might be crucial for understanding organisms. Organicists reject mere adaptationist and selectionist perspectives because of their insufficient explanation of the autonomy of living beings for regulating their processes in relation to environmental conditions [12]. On the contrary, FEP advocates rely on the power of formalism to incorporate any regulation whatsoever. Such disagreement becomes transparent in the conceptual differences regarding what counts as “surprising” for an organism. Whilst organicists stress that surprising events need not always be maladaptive for organisms, Friston and colleagues answer by nuancing the term “surprise”, making it contingent upon the context in which the FEP framework works and the “temporal depth” and “epistemic affordances” of different kinds of sentient systems [13].

Importantly, the FEP framework relies on representations of the external world via a generalized inferential picture of cognition. Obviously, the system needs to start sampling the environment according to certain prior beliefs—equivalent to setting initial conditions—that allegedly recapitulate environmental patterns, but do not in themselves enable efficient inference; the FEP needs to be invoked for that. However, it might also happen that, even if formalism by construction converges on a course of action, trajectory or policy, it fails to reproduce the actual system behavior because of unforeseeable changes in its phase space. Last but not least, since the FEP framework admits the duality of information geometries in its probability distributions —intrinsic (for physical states) and extrinsic (referring to belief states)—how should such probabilities be consistently interpreted? Are interpretations of probabilities as physical propensities compatible with interpretations of probabilities as cognitive inferences at all levels? If, as Colombo and Wright argue [12], Friston interprets probabilities involved in the FEP as objective features of systems [17], a clash seems to emerge between (physical) propensities as causal tendencies that should be asymmetric, like causal relationships, and epistemic probabilities (cognitive inferences) that should not. Conditional probabilities can be reversed—in this sense, they are symmetric, even if reversed conditional probability might be very different from the initial probability. However, cause and effect relationships cannot in general be reversed. Convergence of the information geometry’s dual aspects onto a single objective probability may thus not be that straightforward. (The degree of identification between the two aspects in information geometry is far from clear, as Kiefer has recently pointed out [47]. Obviously, if one-to-one correspondence is relaxed, MM is in trouble as an ontology for the FEP. As a matter of fact, the identification of physical and cognitive aspects in a single physical concept is becoming fashionable in some theories about consciousness. The most well-known example is Integrated Information Theory (IIT)—see e.g., [3], where phenomenological axioms are straightforwardly translated into mathematical axioms. Yet, such a procedure is defective without further clarification [48].)

## 5. MM as the Philosophical Position Adopted by FEP Theorists

Understandably, critiques reviewed in the last section prompt discussion on the FEP’s philosophical interpretations found at the end of [13]. In that work, the authors support MM and establish connections with extant theories regarding the relationship between mind and matter, such as neutral monism, panprotopsychism, dual-aspect theories, and physicalism. As they note from the outset, “[t]he deeper philosophical issue of sentience speaks to the hard problem of tying down quantitative experience or subjective experience within the information geometry afforded by the Markov blanket construction”. This section picks up that gauntlet and engages in the philosophical fray to assess the benefits and shortcomings of MM, and then discusses other, more pressing philosophical issues in Section 6 and Section 7.

### 5.1. A Dual-Aspect Dynamic

To recap, due to the Markov blanket factorization used for the system and from the existence of $p\left(\left\{\mu_{j}\right\}|\left\{s_{k}\right\},\left\{a_{l}\right\}\right)$ [13], assume the existence of a generative model of the external world. Let us call it $p\prime \left(\left\{\eta_{i}\right\}|\left\{\mu_{j}\right\},\;\left\{s_{k}\right\},\left\{a_{l}\right\}\right)$, as the best guess for $q\left(\left\{\eta_{i}\right\}\right)$, i.e., beliefs about the environment encoded in the particular (given) system’s state $x=\left\{\mu_{j}\right\}\;U\;\left\{s_{k}\right\}\;U\;\left\{a_{l}\right\}$. (For details on how such probability densities relate to one another see e.g., [20,49].) p and $p^{\prime }$ stand in 1:1 correspondence, thus defining information geometries for both in such a way that the “[t]he extrinsic geometry [of $p^{\prime }$] is conjugate to the intrinsic geometry [of p] but measures distances between beliefs. Both are measurable, and both supervene on the same Langevin dynamics”. (The Markov blanket ensures that both geometries coincide and allows for the remarkable observation that the non-equilibrium steady state density towards which the system evolves (in terms of its intrinsic geometry) can be interpreted as a statistical or generative model in terms of its extrinsic geometry—i.e., as the joint probability density over systemic or particular states (internal and blanket states) and external states.) Internal states possess a dual aspect information geometry that does not, in and of itself, give a system mental states and consciousness; instead, it only confers computational properties. However, even though physical and computational properties are not identical, “the extrinsic information geometry is ultimately reducible to the intrinsic information geometry (and the other way around), in the sense that there is a necessary link between them”. (As a consequence, this ultimate reduction can only mean embracing a compatibilist view on the problem of free will.) One is thus permitted to express system dynamics “in terms of forces supplied by the extrinsic, belief-based information geometry”, because “[t]he forces that engender our physical dynamics can either be expressed as thermodynamic forces or as self-evidencing; in virtue of the extrinsic information geometry supplied by variational free energy”. The latter “is a feature of an extrinsic information geometry induced by beliefs encoded by internal states that have an intrinsic information geometry” [13].

Complexity becomes a matter of temporal depth, with time scales as grey zones—without clear-cut thresholds—which nevertheless allow us to distinguish between different kinds of systems. Self-organization is simply implied by quick convergence on minimal free energy whilst complex adaptive systems present an adaptive response to changes in their environment that “will look as if they are selecting their long-term actions on the basis of an expected free energy”. Humans, for instance, will possess “deep generative models that see far into the future; enabling a move from homoeostasis to allostasis and, effectively, the capacity to select courses of action that consider long term consequences”. Inasmuch as the system does have temporal depth, it needs to implement estimates of the changing environment through a changing generative model. Hence, within the FEP framework, intentional behavior looks like uncertainty resolving, information seeking, epistemic foraging [13]. However, this is just the dual aspect of underlying physical dynamics spawned by the Markov blanket’s partition of system and environment.

### 5.2. MM Rejects Dualism

Friston and colleagues adopt MM as the better metaphysical position to support the FEP framework. MM can be summed up in these two affirmations: “(1) Fundamentally, there is only one type of thing and only one type of irreducible property … (2) All systems possessing a Markov blanket have properties that are relevant for understanding the mind and consciousness: if such systems have mental properties, then they have them partly by virtue of possessing a Markov blanket” [13]. Of course, one might already question the meaning of “partly” in the second claim, which features an allegedly monist position, and whether such a clarification is compatible with the first claim.

The crucial argument draws on the inter-theoretical relationship between the two possible (considerations of) information geometries (corresponding to p and $p^{\prime }$) and between the properties involved. “Since the dynamics that can be described with reference to these properties can equivalently be described without regarding internal states as representations of probability distributions, there is a sense in which both perspectives are reducible to one another. Hence, the dual information geometry itself does not entail property dualism”. The only possible interpretation of such a claim is that the “dual aspect” of information geometry is merely epistemic, by no means ontic. Moreover, supposedly irreducible mental properties would have to be largely independent of extrinsic information geometry, in blatant contradiction with FEP formalism [13].

FEP theorists acknowledge that FEP formalism could be deemed compatible with (property) dualism but—were that the case—the latter could not explain the existence of minds and consciousness by leveraging properties entailed by the existence of a Markov blanket, even if such properties are identified with protophenomenal properties [13]. Property dualism, according to these authors, demands causes and explanations that differ from those that MM is able to supply. However, strong evidence for the realism demanded by property dualism seems to be lacking—with MM as the best metaphysical option. We will turn back to this issue when reflecting on the implicit assumptions that undergird the FEP framework in the next sections.

### 5.3. MM in Favor of Reductive Materialism

Not surprisingly, MM fares better in the company of metaphysical accounts from neutral monists. The former might be considered a specific version of the latter “in which basic entities are intrinsically neither mental nor physical”—the last two qualifications being two conjugate ways of describing existing stuff. Nevertheless, [13] warn against a surreptitious realist interpretation of both descriptions in terms of extrinsic information geometry, which could sneak property dualism in through the back door. MM’s metaphysical commitment becomes crystal clear when stating that it “is similar to dual-aspect monism… in that it entails that one and the same thing (i.e., internal states of a system possessing a Markov blanket) can be viewed from two perspectives”. But differences do exist between MM and dual-aspect (neutral) monism: “In order to count as a dual-aspect monism, these two perspectives would have to be mutually irreducible … we are skeptical that this would be a coherent interpretation of the dual information geometry”. Briefly put, differences between MM and dual-aspect monism arise out of the purported reducibility of extrinsic to intrinsic information geometry in MM.

What other options are available? MM can ground other versions of reductive materialism, namely a physicalist interpretation. One might thus assume that such choice is motivated by physics’ (alleged) ability to explain the dual aspect of the system’s information geometry. In reality, “[t]here are no additional, non-reducible properties, which are necessary to explain the mind and consciousness; between some non-conscious and conscious systems, there is only a gradual difference”. An implicit recourse to sorites paradoxes—arising from vague predicates—thus transpires when claiming vagueness for consciousness: “If consciousness is a vague concept (as suggested by our interpretation of Markovian Monism), then the right structure and functions can be metaphysically sufficient for consciousness, even if adding just a bit of structure and function to any uncontroversially non-conscious system does not make it conscious” [13]. Even though there are still categorical differences that can be described in terms of more high-level properties, e.g., intentionality and computation, such categories stand in the epiphenomenal realm of physicalist interpretations. Friston and colleagues map the distinction between conscious/non-conscious systems to temporally deep/shallow generative models—which, according to them, is vague and just a quantitative issue. Consequently, MM too serves as a foundation for physicalist approaches to consciousness and the mind.

## 6. What Matters Philosophically: The Implicit Made Explicit

The previous section surveyed some philosophical tenets related to MM as the supposedly best ontology for the FEP, while hinting at its potential problems. In this section, I will endeavor to make explicit some FEP assumptions as a comprehensive scientific program to deal with life phenomena and demonstrate the sense in which said assumptions invalidate the MM perspective. This section aims to show that, contrary to criticisms of the FEP’s attempt at being a GUT on behalf of epistemic pluralism [7], one should cling to scientific reduction as much as possible and, within the conceptual framework set up by the FEP, seek to elucidate the latter’s epistemic advancement. In keeping with this viewpoint, critique of philosophical perspectives and of scientific concepts can more productively come together.

### 6.1. Representation, Probability and Optimization

Assuming the validity of the system/environment partition through a Markov blanket—first circularity—the FEP framework’s central hypothesis is that systems do represent the outer world via their internal states—second circularity. This hypothesis is at the core of the correspondence between p and $p^{\prime }$ and, consequently, of the existence of intrinsic and extrinsic information geometries for the probability distribution of particular states. Obviously, the term “representation” is highly controversial both in cognitive science and contemporary epistemology, (See e.g., [50]. For an overview of the issues surrounding representationalism here, see [20,51,52,53]) but, beyond ongoing debates, it is relevant to consider the emergence of internal representation itself in some living systems during the course of evolution. In other words, if representation is such a crucial concept for the FEP’s performance, discussion of it should go beyond vague descriptions. One could also just speak of information processing; however, internal representation seems closely related to cognition as a product of biological processes. Yet, cognition—or at least some sophisticated level of cognition—is made possible by the presence of a nervous system; to simply equate cognition or internal representation with the ability to generate a response to environmental stimulus in the Markov blanket stretches the term as to make it meaningless, according to some authors ([54], pp. 206–207). If the FEP strives to encompass the evolutionary framework too, the emergence of representation in living systems should not be taken for granted.

The FEP’s internal representations take on a probabilistic nature. Information is encoded and computed through probability density functions or approximations thereof: “This treatment assumes that the system’s state and structure encode an implicit and probabilistic model of the environment” ([18], p. 70). Needless to say, throughout the history of philosophy of science, the very concept of probability oscillates between extreme objectivity and extreme subjectivity, not to mention its essential role in Quantum Mechanics and related interpretations, where probability is hotly debated as a primitive or derivative concept. The point here is not realism or antirealism of scientific theories, as briefly mentioned in Section 4, but the internal coherence of the concept of probability implicitly endorsed by the FEP. To wit, on the one hand, the probabilistic representation of the environment underlies the system’s cascade of Bayesian inferences, making it deeply “subjective” in the sense of system-dependent; on the other hand, probabilities must be objective if the FEP’s overall framework means something and if physicalism and the MM interpretation can be meaningfully held. Hence, it seems that the use of the terms representation and probabilistic representation have not been sufficiently thought through within MM. From its inception, the FEP’s key concepts contain an implicit duality between physics and internal representations that is fully disclosed in the dual information geometry of the probability distributions.

Sure enough, the FEP framework exploits the powerful feature of variational methods to tackle statistical inference problems as optimization problems ([12], n. 6). Nevertheless, the shadow of secondary circularity looms large throughout the whole procedure inasmuch as physical interactions and internal representations remain conceptually entangled whenever inference (epistemic) is reduced to optimization (ontic). Let me illustrate this point with a clarifying example. For the whole variational procedure to make sense and the risk of infinite regress to be ruled out, one needs to begin mathematically with some priors that are usually chosen using random values. Of course, one legitimate procedure for solving the optimization problem does not and cannot provide a single cue about the origin of priors in the actual inference process. It is one thing, as a scientist, to seed a well-defined optimization procedure with random priors to hopefully mimic some aspect of reality; another very different thing concerns whether and how inference works in actual natural systems. (In the FEP framework, one may always say that priors are just empirical priors. For example, yesterday’s posterior is today’s prior. This means priors always inherit from somewhere else. This can be seen structurally in hierarchical generative models, where the priors at one level can be regarded as likelihoods from the point of view of the level above. But this answer simply begs the ontological question since levels are a priori constructed within each concrete FEP model.) As I will show in the last section, once we reckon the FEP limits, the abovementioned circularities are not necessarily a drawback for the FEP as a scientifically valid framework; they are, though, when FEP is interpreted through MM.

### 6.2. Supervenience, Causality and Individuation

Friston and colleagues’ reference to the emergence and coexistence of a duality, intrinsic and extrinsic, in information geometry is extremely illustrative. Through the free energy minimization procedure, “belief updating and statistical thermodynamics both *supervene* on the same internal manifold”. If “[i]t is tenable to associate physics (in the sense of quantum, statistical and classical) mechanics with the intrinsic information geometry” [13], the logical consequence is to associate extrinsic information geometry with internal beliefs. However, whereas the possibility that statistical thermodynamics may supervene on ultimately physical stuff seems relatively straightforward, the sense in which beliefs also supervene on the same stuff remains much more controversial.

Be that as it may, supervenience is the philosophical concept used to allow for the dual aspect of the systems’ internal basic stuff. As is well known, the explanatory power of supervenience is also highly controversial, though supervenience is usually deemed a good ally of physicalism ([55], sec. 5.4), the grounding pillar of MM. Even though “Bayesian mechanics must still apply, even during the suspension of any coupling with blanket states” [13], one might consider the corresponding extrinsic information geometry as purely remnant. The supervenient mechanics of beliefs need not have causal power since internal dynamics can also be explained without reference to extrinsic information, as physicalism requires. However, from the viewpoint of metaphysical causation supervenience is a descriptive label that still has to come to terms with causes. Here, one may recall the mechanist critique of the FEP: “Filling out the mechanism sketch is what matters: appeals to the mechanism of predictive coding—not FEP—are what provides explanatory depth” [12]. Noticeably, affirming in the extended FEP framework that, “the expected free energy influences policy selection” and “that future or latent outcomes have the potential to influence beliefs about past states” ([29], p. 503) seems to imply that the future influences the past, which does not easily harmonize with a strict, non-retrocausal physicalism.

The hidden problem in the preceding discussion has to do with a lack of explicitness regarding the FEP framework’s implicit assumptions. Such need for explicitness behooves philosophical criticism. In fact, some of the model’s underlying causal hypotheses more or less overtly reveal themselves during the technical resolution of a free-energy minimization. Lastly, since the FEP is formalized by a set of differential equations, well-defined criteria for the establishment of effective boundary conditions are mandatory. Yet, such technical requirements bring us back to the system’s very definition and individuation with the help of its Markov blanket: “This move is crucial for elaborating a physics of sentience, in which physical dynamics entail probabilistic beliefs about something … there would be no quantum or statistical mechanics in the absence of Markov blankets” [13]. These authors are thus bound to a notion of duality or conjugacy required by the Markov blanket partition, but that proviso inevitably begs deeper questions. Namely, what is an individual system? What kind of causality does it entail that allows scientists to speak of systems and Markov blankets?

## 7. What Is an Individual System?

One of organicists’ deepest criticisms of the FEP, as hinted at in Section 4, amounts to the degree of specificity of physical systems. “If historical considerations and lineage matter to understanding organisms and their dynamics, then biological systems should be represented as ‘specific’ and their trajectories as ‘generic.’ Instead … free-energy theorists get it backward: physical systems are ‘generic,’ while their trajectories ‘specific’” ([12,43], Chapter 7). Behind that critique, one may divine a certain mistrust for understanding individual organisms through the FEP alone. Moreover, if living systems are not just optimization processes, but rather “extended critical transitions, always transient toward a continually renewed structure” ([43], p. 162), their individuality may turn out to be unique in the sense of the impossibility of wholly explaining them away by general principles.

Remarkably enough, FEP theorists start off with the observation that some systems maintain their physical integrity, displaying adaptive behavior in a changing environment. Systems are featured as remaining in a non-equilibrium steady state, within a relatively narrow region of all possible states of their initial phase space. If that is the case, Markov blankets can be deemed objective features of nature, separating biological systems’ internal states from those external to them [17]. Then, “by acting on the environment to minimize the free energy of their sensory samples, biological systems would avoid surprising sensory states. If they avoid surprising sensory states, biological systems may attain a homeostatic state; and by selecting actions that attain homeostatic states, biological systems will thereby behave adaptively and preserve their physical integrity” [12].

However, such “generalized homeostasis” ([13], n. 7) rests upon delicate, non-fundamental, stipulations—e.g., contingent upon a concrete range of energy exchange for some timespan (the duration over which the system exists) between the system and the environment. Only within that regime does it make sense to a priori define the relevant degrees of freedom for the system, including a specific Markov blanket parametrizing the system’s interaction with the environment. Yet, is it possible to a priori define the relevant degrees of freedom or states for complex adaptive systems? Such systems can continually increase the diversity of what happens next [45]. How might we define the system’s relevant degrees of freedom in fundamental terms when complex adaptive systems’ autopoiesis remains controversial [56]? Even with an effort in the FEP literature to link priors to phenotypes [49] in the FEP’s framework, existence just means the attainment of a steady state in a system without explaining its evolutionary history. However, such a perspective is flawed as it cannot encompass a system’s changing phase space or non-steady probability densities related to the system’s states, both of which might prove essential in understanding life phenomena.

In other words, the FEP suffers from circularity because it a priori assumes conditions that are to be maintained. No novelty can emerge without further assumptions within this model. For Friston and colleagues, “A particle or person is never ‘off’ their manifold—they just occupy states that are more or less likely, given the kind of thing they are” [13]. Still, life phenomena deserve further explanation, namely why does identity remain in an ever-changing universe? Is identity merely epistemic or contextually ontic? If it is the latter, why do such contexts allow for the emergence of individual systems? The FEP framework simply takes as given that particles, persons or any system whatsoever are “the kind of thing they are”. That may be enough for science, but it is far from enough for philosophy.

In short, the contextuality of system/environment separation always lurks behind any would-be fundamental theory of life and sentience, and the FEP framework is not immune to that. The FEP deserves credit for showing how “action and perception look as if they are minimizing a particular entropy” [13]. Yet, it is only too well-known that physics still faces a fragmentary landscape regarding the application of variational principles to free energy minimization or entropy production. What is more, a pragmatic approach seems the only way forward as far as “the empirical and numerical evidence appears to suggest that there is no universal entropy production functional that is maximized in all problems” ([57], p. 19).

Nonetheless, in as far as its limitations are acknowledged, such circularities need not result in a drawback for the theory. The FEP might be circular, but, in moving around a circle whose foundations are accepted, it provides epistemic gain through a unified principle featuring homeostatic-like phases in living systems’ complex phase space. There, any system characterized as possessing a Markov blanket “can be cast as performing some elemental form of inference—and possessing an implicit generative model” [13]. As a good ally to complexity sciences, FEP formalism introduces a specific frame in which meaning itself might emerge as intimately related to systems’ survival and natural selection: “Events that happen to an organism mean something to that organism if those events affect its well-being or reproductive abilities. In short, the meaning of an event is what tells one how to respond to it … This focus on fitness is one way I can make sense of the notion of meaning and apply it to biological information-processing systems” ([58], p. 184). Even so, the FEP’s circularities remind us of its contextually limited explanatory power; in complex systems, “who or what actually perceives the meaning of situations so as to take appropriate actions? This is essentially the question of what constitutes consciousness or self-awareness in living systems … [This is one] among the most profound mysteries in complex systems and in science in general. Although this mystery has been the subject of many books of science and philosophy, it has not yet been completely explained to anyone’s satisfaction” ([58], p. 184). Hence, the objections to MM raised here also impinge on scientific attempts at using the FEP to understand the emergence of inference, representation or meaning. More specifically, the FEP is explanatory as a way of making predictions “as if” an extant physical system behaves with a dual geometry. However, the FEP does not explain the existence of inference, representation or meaning in nature.

## 8. Conclusions

Throughout the previous sections, this paper has submitted the FEP framework (Section 2) to profound philosophical inspection. It has nuanced previous examination of the FEP from Colombo and Wright [12] in the following sense: whereas general criticisms of the FEP as a GUT in terms of epistemic pluralism seem misplaced, since they tend to cut off the spirit of scientific reduction and theoretical scientific progress (Section 3), other criticisms from mechanists and organicists stand and ask for deeper metaphysical scrutiny (Section 4). The opportunity for such scrutiny comes on the back of FEP theorists embracing philosophy in a recent publication [13], where they defend so-called Markovian Monism (MM) (Section 5). This is good news for interdisciplinary work between science and philosophy without explicitly requiring scientific explanatory pluralism.

MM was presented and reviewed in this paper by making explicit the FEP’s implicit assumptions, which Friston and colleagues insufficiently disclosed (Section 6). In particular, MM does not tackle the emergence of representation through evolution, taking for granted some living systems’ specific cognitive ability. More importantly, MM veils the assumptions behind separating system and environment via a Markov blanket. Such drawbacks heavily stymie a FEP interpretation consistent with MM. Nonetheless, inasmuch as the FEP itself is not committed to MM, such critiques do not invalidate the epistemic advancement that the FEP framework provides. Even though the FEP may incur in circular reasoning, this drawback simply weighs upon its Markovian monist interpretation, and not necessarily on its value as a generalized heuristic principle for better characterization of homeostatic phases in already extant biological systems (Section 7). The FEP’s recourse to Markov blankets induces circular causalities between living systems and the environment, which, freed from MM interpretations, can go beyond circular reasoning.

## Data Availability

No applicable.

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
