# Peer review of "The Free Energy Principle: Good Science and Questionable Philosophy in a Grand Unifying Theory"

_entropy, 2021, doi:10.3390/e23020238_

Round 1

Reviewer 1 Report

Major revisions

This manuscript is not suitable for publication in Entropy in its current state, but it might be pending some fairly substantial revisions – and some much-needed trimming. The main contribution of this paper, if I’ve understood the authors’ aim, is its attempt to discuss a number of philosophical interpretations of the mind-body relation that may or may not be consistent with the free-energy principle (FEP). The paper, in particular, argues that the favored philosophical interpretation of the FEP, Markovian monism (MM), encounters some problems. This theme is broadly relevant for the readership of this journal.

The paper has potential. However, as it stands now, the argument is a bit all over the place. The paper is currently also rather unpolished (with missing elements, such as citations with no associated references) and encounters some fairly significant problems in terms of the discussion of the FEP. I think several aspects of the discussion, especially more technical ones, could be improved, and the focus of the paper made much clearer.

1. Paper tries to cover too much ground

The paper tries to cover too much ground and in so doing is restricted to a fairly superficial analysis of several topics. The paper tries to do some things that other papers do much better, with a devoted analysis. To be more precise, the discussion of realism versus instrumentalism in this paper is too short and partial to be very useful, and other papers exist that do a better job, e.g., (Andrews, 2020; Bruineberg, Dolega, Dewhurst, & Baltieri, 2020; Ramstead, Friston, & Hipólito, 2020; Ramstead, Hesp, et al., 2020; Ramstead, Kirchhoff, & Friston, 2019). The issues surrounding realism and instrumentalism are not discussed philosophically, for instance; and these issues are serious enough to figure in the discussion. Specifically, the question of the reality of Markov blankets has been addressed much more extensively and with greater mathematical precision in (Bruineberg et al., 2020). More excellent commentary on the question of realism versus instrumentalism can be found in (Andrews, 2020). The questions surrounding representationalism have been addressed in paper-length treatments that have already been published elsewhere (Constant, Clark, & Friston, 2019; Gładziejewski, 2016; Ramstead, Friston, et al., 2020; Ramstead, Kirchhoff, et al., 2019; Williams, 2017). Many of these papers are not cited in the manuscript.

Each of the critical philosophical subsections (5, 6, and 7) could be a self-contained paper. I recommend submitting them as such, with expanded discussion. Section 5 is just commentary on some (not very good) philosophical commentary, and doesn’t contribute all that much to the argument on offer about MM. I would recommend cutting it, especially since the analysis by Colombo and colleagues is highly questionable. The substantial contribution of the paper is found in section 6, which should be substantiated more in my view.

One major problem with the paper as it stands is that the FEP itself is not committed to MM. Indeed, MM is just one possible interpretation, albeit one favored by the authors. This is mentioned in the article but should be made much clearer. Moreover, the positions themselves (dualism, dual-aspect monism, etc.) should be discussed and not assumed.

2. Technical accuracy of description of FEP is lacking

The discussion of the FEP is fair but lacking in some central respects. I think the paper could be significantly improved, and the central argument strengthened, by appealing to more of the formal resources of the framework. This is relevant because the authors’ argument explicitly aims to follow from an analysis of the formalism.

Although I do not necessarily expect the author to fully discuss all technical details, this is what the free-energy principle does—discussion based on Friston (2020); Friston, Wiese, and Hobson (2020); Ramstead, Friston, et al. (2020).

The explanatory core of the FEP rests on the dual information geometry of self-organizing systems with a set of characteristic states that possesses a Markov blanket and a nonequilibrium steady state density (Friston, 2020). The FEP allows us to systematically associate semantic contents to physical states and their dynamics.

Basically, the idea is that the FEP allows us to describe a system in one of two mathematically conjugate or equivalent ways – an equivalence that rests on the fact that the system being considered has a Markov blanket and exists at nonequilibrium steady state. The Markov blanket is a set of variables or states that individuates system by identifying those states that mediate interactions between the system and its embedding environment.

The Markov blanket is itself defined by the absence of certain connections: internal states do not cause sensory states and external states do not cause active states. It is this absence of connectivity that defines the Markov blanket – and is that upon which rests much of the mathematical heavy lifting. Indeed, the core intuition behind the free-energy principle is that there are two ways of describing the flow of a system’s states that turn out to be equivalent. One can always describe any dynamical system in terms of its phase space, where every dimension of this space corresponds to a variable in the system; such that a point in this space (which assigns a value along each dimension) represents the instantaneous state of the system, and that a trajectory corresponds to the flow of states over time. This description of the system in terms of trajectories in phase space over time is the system’s ‘intrinsic’ information geometry.

The FEP allows us to say that, so long as the Markov blanket is in play, then an additional – and, crucially, mathematically conjugate – information geometry can be defined for the system. This ‘extrinsic’ geometry is a trajectory not in a phase space of states, but over a statistical manifold. A statistical manifold is a space in which the coordinates are the sufficient statistics of families of probabilities densities; where sufficient statistics are the numbers that define a given probability distribution. E.g., the sufficient statistics of a Gaussian or normal distribution are its mean and variance; given these, it is possible reconstruct the (infinite number of points) of that distribution.

What the free-energy principle really says is that so long as a Markov blanket is in play, gradient flow on variational free-energy in phase space over time is equivalent to gradient flow on variational free-energy in a statistical manifold, where the sufficient statistics are of beliefs or probability distributions about external states.

The Markov blanket – because it implements conditional independence, since it ensures the systematic absence of certain causal relations – ensures that both geometries coincide, and licences a spectacular observation: namely, that the nonequilibrium steady state density towards which the system evolves (in terms of its intrinsic geometry) can be interpreted as a statistical or generative model in terms of its extrinsic geometry – i.e., as the joint probability density over systemic or particular states (internal and blanket states) and external states.

This is the core intuition behind the FEP. In plain English, then, if the FEP is applicable to some system, then it is justified to speak of the system as if it had mental states, where “as if” doesn’t refer to scientific instrumentalism but rather to the duality of perspectives just explained.

I note that the authors’ discussion of Markov blankets is inaccurate. E.g., they write, “Since sensory states do not directly depend on the system’s active and internal states, active and internal states will look as if they are trying to minimize exactly the same quantity; namely, the surprisal of states that constitute the system.” This is mathematically incorrect. Sensory and active states can influence each other, the only thing precluded is external state influence on active states and internal state influence on sensory states. With respect to this paper’s aims, it would be easier to explain Markov blankets in terms of sparsity of coupling: some connections are precluded, namely, internal states don’t affect sensory states and external states don’t affect active states. Or again: “In the context of FEP, this optimization involves changes only to internal parameters (Colombo and Wright 2018).” This is also wrong, since action selection is premised on the same formulation; moreover, it is not just parameter estimation but also state estimation that is finessed through free-energy minimization.

3. More technical shortcomings

Some central constructs of the free-energy approach should be better explained. Terms like ‘surprisal’, ‘variational free-energy’, ‘generative model’, ‘complexity’, ‘temporal depth’, ‘information geometry’, ‘Sorites paradoxes’, etc., are introduced without definitions or discussion. In particular, ergodicity is mentioned as a problem without discussion. On the most recent construction, ergodicity is not a problem for the FEP framework because the FEP framework only assumes local ergodicity: relative to the parts that make it up, each whole looks stationary. For instance, for the entire lifecycle of a cell in my heart, my heart’s overall phenotype will have remained stable. So, while life might not be ergodic globally, it is locally ergodic enough to, e.g., be measured as having locally stable properties. Analogously, the Earth is not flat, but it is locally flat enough for us to build skyscrapers. See (Friston, 2020)

What’s missing from the picture in this paper is a discussion of how the NESS density/generative model p that defines the variational free-energy, and how these are related to the proposal or variational distributions q that are used to infer the true posterior. These are mischaracterized at several points in the paper. Please see the discussion of semantics and representation in (Ramstead, Friston, et al., 2020; Ramstead, Kirchhoff, et al., 2019).

The discussion of information geometry is very incomplete, and in fact comes out of nowhere. Information geometry is not explained and the mechanics of information geometry under the FEP are not discussed. As a consequence, the discussion of how semantics gets into the picture is confused. Internal states do not stand for themselves. The idea is not that of “measuring differences between probabilities of internal (physical) states or of beliefs (about external states).” Again, in consequence, the discussion in section 5.1. is off the mark. The generative model and the recognition density are not the same, and since they are connected to how the variational free-energy itself is defined, the distinction should be discussed in more detail.

Footnote 3 misses some crucial points about free-energy minimization. It is not the case that free-energy gradients need to be instantiated explicitly, it is just the case that the system’s flow it its state and belief space minimize a free-energy functional; again, see (Ramstead, Friston, et al., 2020; Ramstead, Kirchhoff, et al., 2019).

4. Consciousness

Consciousness is not defined in this paper. Friston and colleagues do not mean phenomenal consciousness, they mean something like awareness or sentience, i.e., the capacity or ability to respond adaptively and appropriately to the demands of a situation. Technically the FEP is presented as a theory of sentient systems (Ramstead, Constant, Badcock, & Friston, 2019).

References

Andrews, M. (2020). The Math is not the Territory: Navigating the Free Energy Principle.
Bruineberg, J., Dolega, K., Dewhurst, J., & Baltieri, M. (2020). The Emperor’s New Markov Blankets.
Constant, A., Clark, A., & Friston, K. J. (2019). Representation wars: Enacting an armistice through active inference. PhilSci Archive (preprint).
Friston, K. J. (2020). A free energy principle for a particular physics.
Friston, K. J., Wiese, W., & Hobson, J. A. (2020). Sentience and the origins of consciousness: From Cartesian duality to Markovian monism. Entropy, 22(5), 516.
Gładziejewski, P. (2016). Predictive coding and representationalism. Synthese, 193(2), 559-582.
Ramstead, M. J., Constant, A., Badcock, P. B., & Friston, K. J. (2019). Variational ecology and the physics of sentient systems. Physics of life Reviews.
Ramstead, M. J., Friston, K. J., & Hipólito, I. (2020). Is the free-energy principle a formal theory of semantics? From variational density dynamics to neural and phenotypic representations. Entropy.
Ramstead, M. J., Hesp, C., Tschantz, A., Smith, R., Constant, A., & Friston, K. (2020). Neural and phenotypic representation under the free-energy principle. Neuroscience & Biobehavioral Reviews.
Ramstead, M. J., Kirchhoff, M. D., & Friston, K. J. (2019). A tale of two densities: active inference is enactive inference. Adaptive Behavior, 1059712319862774.
Williams, D. (2017). Predictive processing and the representation wars. Minds and Machines, 1-32.

Reviewer 2 Report

This is a well-written and insightful discussion the strengths and weaknesses of the FEP.

However, revisions are needed before it is ready as a contribution to the literature.

I would change the title to say "questionable", rather than "bad".

The process theory of Active Inference (AI) should be discussed in addition to the FEP.

Active Inference: A Process Theory - PubMed (nih.gov)NECO_a_00912 (mitpressjournals.org)

The Markov blankets of life: autonomy, active inference and the free energy principle - PubMed (nih.gov)

AI takes us from generalized homeostasis/autopoiesis to allostasis, which may address some of the authors concerns regarding the dynamic (and self-organized critical) nature of life.

I would also mention the following three preprints, which discuss relevant issues including circularity, falsifiability, ontological status of Markov blankets, and representationalism.

The Math is not the Territory: Navigating the Free Energy Principle - PhilSci-Archive (pitt.edu)

The Emperor’s New Markov Blankets - PhilSci-Archive (pitt.edu)

Free-Energy Principle, Computationalism and Realism: a Tragedy - PhilSci-Archive (pitt.edu) 

Regarding consciousness, it might be worth briefly commenting on previous attempts at discussing that subject within the context of FEP-AI, some of which have fewer panprotopsychist implications.

E.g.

A mathematical model of embodied consciousness - PubMed (nih.gov)

The predictive global neuronal workspace: A formal active inference model of visual consciousness - PubMed (nih.gov)

Frontiers | An Integrated World Modeling Theory (IWMT) of Consciousness: Combining Integrated Information and Global Neuronal Workspace Theories With the Free Energy Principle and Active Inference Framework; Toward Solving the Hard Problem and Characterizing Agentic Causation | Artificial Intelligence (frontiersin.org)

Reviewer 3 Report

Please see attached PDF for comments.

Round 2

Reviewer 2 Report

Having read through the revised version of the manuscript, I feel that all of my concerns have been more than adequately addressed. This paper not only provides a cogent summary of the FEP that will be useful to the uninitiated, but it also highlights some of the ongoing challenges currently being debated within this evolving paradigm. While ideally I would have liked to have seen more engagement with the Active Inference framework and particular theories of consciousness within FEP-AI, I think that's something that can be saved for a future article, which I hope the author writes in the not distant future.

Author Response

I am grateful to referee 2 for his or her comments and, especially, for the suggestion on writing a future article on the Active Inference framework and particular theories of consciousness within FEP-AI.

Author Response

I thank the referee for his/her comments. Please, see my answers in the attached file.
